# Document Classification for COVID-19 Literature

**Bernal Jiménez Gutiérrez, Juncheng Zeng, Dongdong Zhang, Ping Zhang, Yu Su**

The Ohio State University

`{jimenezgutierrez.1,zeng.671,zhang.11069,`
`zhang.10631,su.809}@osu.edu`

## Abstract

The global pandemic has made it more important than ever to quickly and accurately retrieve relevant scientific literature for effective consumption by researchers in a wide range of fields. We provide an analysis of several multi-label document classification models on the LitCovid dataset, a growing collection of 8,000 research papers regarding the novel 2019 coronavirus. We find that pre-trained language models fine-tuned on this dataset outperform all other baselines and that the BioBERT and novel Longformer models surpass all others with almost equivalent micro-F1 and accuracy scores of around 81% and 69% on the test set. We evaluate the data efficiency and generalizability of these models as essential features of any system prepared to deal with an urgent situation like the current health crisis. Finally, we explore 50 errors made by the best performing models on LitCovid documents and find that they often (1) correlate certain labels too closely together and (2) fail to focus on discriminative sections of the articles; both of which are important issues to address in future work. Both data and code are available on GitHub [1].

## 1 Introduction

The COVID-19 pandemic has made it a global priority for research on the subject to be developed at unprecedented rates. Researchers in a wide variety of fields, from clinicians to epidemiologists to policy makers, must all have effective access to the most up to date publications in their respective areas. Automated document classification can play an important role in organizing the stream of articles by fields and topics to facilitate the search process and speed up research efforts.

To explore how document classification models can help organize COVID-19 research papers, we

use the LitCovid dataset (Chen et al., 2020), a collection of 8,000 newly released scientific papers compiled by the NIH to facilitate access to the literature on all aspects of the virus. This dataset is updated daily and every new article is manually assigned one or more of the following 8 categories: *General*, *Transmission Dynamics (Transmission)*, *Treatment*, *Case Report*, *Epidemic Forecasting (Forecasting)*, *Prevention*, *Mechanism* and *Diagnosis*. We leverage these annotations and the articles made available by LitCovid to compile a timely new dataset for multi-label document classification.

Apart from addressing the pressing needs of the pandemic, this dataset also offers an interesting document classification dataset which spans different biomedical specialities while sharing one overarching topic. This setting is distinct from other biomedical document classification datasets which tend to exclusively distinguish between biomedical topics such as hallmarks of cancer (Baker et al., 2016), chemical exposure methods (Baker, 2017) or diagnosis codes (Du et al., 2019). The dataset's shared focus on the COVID-19 pandemic also sets it apart from open-domain datasets and academic paper classification datasets such as IMDB or the aRxiv Academic Paper Dataset (AAPD) (Yang et al., 2018) in which no shared topic can be found in most of the documents, and it poses unique challenges for document classification models.

We evaluate a number of models on the LitCovid dataset and find that fine-tuning pre-trained language models yields higher performance than traditional machine learning approaches and neural models such as LSTMs (Adhikari et al., 2019b; Kim, 2014; Liu et al., 2017). We also notice that BioBERT (Lee et al., 2019), a BERT model pretrained on the original corpus for BERT plus a large set of PubMed articles, performed slightly better than the original BERT base model. We also observe that the novel Longformer (Beltagy et al.,

---

[1] `https://github.com/dki-lab/covid19-classification`

|                   | LitCovid | CORD-19 Test |
|-------------------|----------|--------------|
| # of Classes      | 8        | 8            |
| # of Articles     | 8,002    | 100          |
| Avg. sentences    | 51       | 109          |
| Avg. tokens       | 1,221    | 2861         |
| Total # of tokens | 9,771,284| 286,065      |

Table 1: Dataset statistics for the *LitCovid* and *Test CORD-19* Datasets.

| Class        | LitCovid | CORD-19 Set |
|--------------|----------|-------------|
| Prevention   | 3807     | 12          |
| Treatment    | 2149     | 20          |
| Diagnosis    | 1570     | 25          |
| Mechanism    | 1199     | 70          |
| Case Report  | 621      | 2           |
| Transmission | 455      | 6           |
| General      | 222      | 7           |
| Forecasting  | 205      | 2           |

Table 2: Number of documents in each category for the *LitCovid* and *CORD-19 Test Datasets*.

2020) model, which allows for processing longer sequences, matches BioBERT's performance when 1024 subwords are used instead of 512, the maximum for BERT models.

We then explore the data efficiency and generalizability of these models as crucial aspects to address for document classification to become a useful tool against outbreaks like this one. Finally, we discuss some issues found in our error analysis such as current models often (1) correlating certain categories too closely with each other and (2) failing to focus on discriminative sections of a document and get distracted by introductory text about COVID-19, which suggest venues for future improvement.

## 2 Datasets

In this section, we describe the LitCovid dataset in more detail and briefly introduce the CORD-19 dataset which we sampled to create a small test set to evaluate model generalizability.

### 2.1 LitCovid

The LitCovid dataset is a collection of recently published PubMed articles which are directly related to the 2019 novel Coronavirus. The dataset contains upwards of 14,000 articles and approximately 2,000 new articles are added every week, making it a comprehensive resource for keeping researchers up to date with the current COVID-19 crisis.

For a large portion of the articles in LitCovid, either the full article or at least the abstract can be downloaded directly from their website. For our document classification dataset, we select 8,002 from the original 14,000+ articles which contain full texts or abstracts. As seen in table 1, these selected articles contain on average approximately 51 sentences and 1,200 tokens, reflecting the roughly even split between abstracts and full articles we observe from inspection.

Each article in LitCovid is assigned one or more of the following 8 topic labels: *Prevention*, *Treatment*, *Diagnosis*, *Mechanism*, *Case Report*, *Transmission*, *Forecasting* and *General*. Even though

every article in the corpus can be labelled with multiple tags, most articles, around 76%, contain only one label. Table 2 shows the label distribution for the subset of LitCovid which is used in the present work. We note that there is a large class imbalance, with the most frequently occurring label appearing almost 20 times as much as the least frequent one. We split the LitCovid dataset into train, dev, test with the ratio 7:1:2.

### 2.2 CORD-19

The COVID-19 Open Research Dataset (CORD-19) (Wang et al., 2020) was one of the earliest datasets released to facilitate cooperation between the computing community and the many relevant actors of the COVID-19 pandemic. It consists of approximately 60,000 papers related to COVID-19 and similar coronaviruses such as SARS and MERS since the SARS epidemic of 2002. Due to their differences in scope, this dataset shares only around 1,200 articles with the LitCovid dataset.

In order to test how our models generalize to a different setting, we asked biomedical experts to label a small set of 100 articles found only in CORD-19. Each article was labelled independently by two annotators. For articles which received two different annotations (around 15%), a third annotator broke ties. Table 1 shows the statistics of this small set and Table 2 shows its category distribution.

## 3 Models

In the following section we provide a brief description of each model and the implementations used. We use micro-F1 (F1) and accuracy (Acc.) as our evaluation metrics, as done in (Adhikari et al., 2019a). All reproducibility information can be found in Appendix A.

### 3.1 Traditional Machine Learning Models

To compare with simpler but competitive traditional baselines we use the default scikit-learn (Pe-

| Model | Dev Set | | Test Set | |
|---|---|---|---|---|
| | Acc. | F1 | Acc. | F1 |
| **LR** | 53.3 | 67.5 | 58.5 | 72.2 |
| **SVM** | 58.8 | 72.4 | 62.6 | 76.0 |
| **LSTM** | 57.7 ±0.7 | 75.8 ±0.5 | 59.1 ±1.3 | 76.1 ±0.5 |
| **LSTM$_{reg}$** | 59.4 ±2.4 | 74.6 ±1.2 | 61.7 ±1.9 | 75.9 ±1.2 |
| **KimCNN** | 59.3 ±1.1 | 75.7 ±0.4 | 61.0 ±0.1 | 76.2 ±0.2 |
| **XML-CNN** | 61.9 ±1.0 | 77.2 ±0.3 | 64.6 ±0.4 | 77.9 ±0.3 |
| **BERT$_{base}$** | 66.1 ±1.3 | 79.1 ±0.1 | 68.1 ±0.9 | 80.6 ±0.2 |
| **BERT$_{large}$** | 66.4 ±0.5 | 79.0 ±0.7 | 68.1 ±1.1 | 79.5 ±1.2 |
| **Longformer** | 66.7 ±1.1 | 79.9 ±0.5 | 69.2 ±0.2 | 80.7 ±0.7 |
| **BioBERT** | 66.5 ±0.6 | 80.2 ±0.1 | 68.5 ±1.0 | 81.2 ±0.3 |

Table 3: Performance for each model expressed as *mean ± standard deviation* across three training runs.

dregosa et al., 2011) implementation of logistic regression and linear support vector machine (SVM) for multi-label classification which trains one classifier per class using a one-vs-rest scheme. Both models use TF-IDF weighted bag-of-words as input.

## 3.2 Conventional Neural Models

Using Hedwig[2], a document classification toolkit, we evaluate the following models: KimCNN (Kim, 2014), XML-CNN (Liu et al., 2017) as well as an unregularized and a regularized LSTM (Adhikari et al., 2019b). We notice that they all perform similarly and slightly better than traditional methods.

## 3.3 Pre-Trained Language Models

Using the same Hedwig document classification toolkit, we evaluate the performance of DocBERT (Adhikari et al., 2019a) on this task with a few different pre-trained language models. We fine-tune BERT base, BERT large (Devlin et al., 2019) and BioBERT (Lee et al., 2019), a version of BERT base which was further pre-trained on a collection of PubMed articles. We find all BERT models achieve best performance with their highest possible sequence length of 512 subwords. Additionally, we fine-tune the pre-trained Longformer (Beltagy et al., 2020) in the same way and find that it performs best when a maximum sequence length of 1024 is used. As in the original Longformer paper, we use global attention on the [CLS] token for document classification but find that performance improves by around 1% if we use the average of all tokens as input instead of only the [CLS] representation. We hypothesize that this effect can be observed because the LitCovid dataset contains longer documents on average that the Hyperpartisan dataset used in the original Longformer paper.

[2]https://github.com/castorini/hedwig

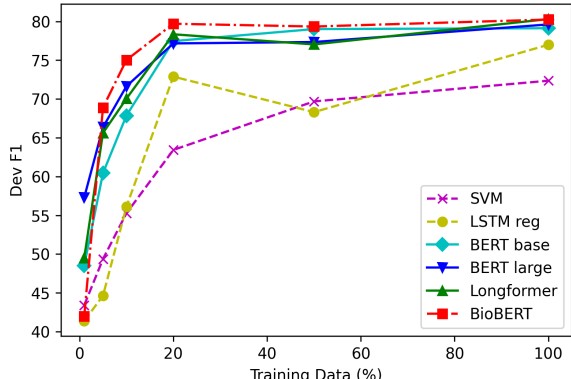

Figure 1: Data efficiency analysis. Pre-trained language models achieve their maximum performance on only 20% of the training data.

We find that all pre-trained language models outperform the previous traditional and neural methods by a sizable margin in both accuracy and micro-F1 score. The best performing models are the Longformer and BioBERT, both achieving a similar micro-F1 score of around 81% on the test set and an accuracy of 69.2% and 68.5% respectively.

## 4 Results & Discussion

In this section, we explore data efficiency, model generalizability and discuss potential ways to improve performance on this task in future work.

## 4.1 Data Efficiency

During a sudden healthcare crisis like this pandemic it is essential for models to obtain useful results as soon as possible. Since labelling biomedical articles is a very time-consuming process, achieving peak performance using less data becomes highly desirable. We thus evaluate the data efficiency of these models by training each of the ones shown in Figure 1 using 1%, 5%, 10%, 20% and 50% of our training data and report the micro-F1 score on the dev set. When selecting the data subsets, we sample each category independently to make sure they are all represented.

We observe that pre-trained models are much more data-efficient than other models and that BioBERT is the most efficient, demonstrating the importance of domain-specific pre-training. We also notice that BioBERT performs worse than other pre-trained models on 1% of the data, suggesting that its pre-training prevents it from leveraging potentially spurious patterns when there is very little data available.

| Article | Label | Prediction |
|---|---|---|
| **Analysis on epidemic situation and spatiotemporal changes of COVID-19 in Anhui.** ... We mapped the spatiotemporal changes of confirmed cases, fitted the epidemic situation by the population growth curve at different stages and took statistical description and analysis of the epidemic situation in Anhui province. | Forecasting | Prevention Forecasting |
| **2019 Novel coronavirus: where we are and what we know.** There is a current worldwide outbreak of a new type of coronavirus (2019-nCoV), which originated from Wuhan in China and has now spread to 17 other countries. ... This paper aggregates and consolidates the virology, epidemiology, clinical management strategies ... In addition, by fitting the number of infections with a single-term exponential model ... | Treatment Mechanism Transmission Forecasting | Prevention Forecasting |
| **Managing Cancer Care During the COVID-19 Pandemic: Agility and Collaboration Toward a Common Goal.** The first confirmed case of coronavirus disease 2019 (COVID-19) in the United States was reported on January 20, 2020, in Snohomish County, Washington. ... | Treatment | Prevention |

Table 4: LitCovid Error Samples. Sentences relevant to the article's category are highlighted with blue and general ones with red.

## 4.2 CORD-19 Generalizability

To effectively respond to this pandemic, experts must not only learn as much as possible about the current virus but also thoroughly understand past epidemics and similar viruses. Thus, it is crucial for models trained on the LitCovid dataset to successfully categorize articles about related epidemics. We therefore evaluate some of our baselines on such articles using our labelled CORD-19 subset. We find that the micro-F1 and accuracy metrics drop by around 10 and 30 points respectively. This massive drop in performance from a minor change in domain indicates that the models have trouble ignoring the overarching COVID-19 topic and isolating relevant signals from each category.

| | Acc. | F1 |
|---|---|---|
| **SVM** | 26.0 | 55.6 |
| **LSTM$_{reg}$** | 31.3 ±2.5 | 62.9 ±2.4 |
| **Longformer** | 37.3 ±4.9 | 66.9 ±2.1 |
| **BioBERT** | 39.7 ±3.1 | 68.1 ±1.3 |

Table 5: Performance on the CORD-19 Test Set expressed as *mean ± standard deviation* across three training runs.

It is interesting to note that *Mechanism* is the only category for which BioBERT performs better in CORD-19 than in LitCovid. This could be due to *Mechanism* articles using technical language and there being enough samples for the models to learn; in contrast with *Forecasting* which also uses specific language but has far fewer training examples. BioBERT's binary F1 scores for each category on both datasets can be found in Appendix B.

## 4.3 Error Analysis

We analyze 50 errors made by both highest scoring BioBERT and the Longformer models on Lit-Covid documents to better understand their performance. We find that 34% of these were annotation errors which our best performing model predicted correctly. We also find that 10% of the errors were nearly impossible to classify using only the text available on LitCovid, and the full articles are needed to make better-informed prediction. From the rest of the errors we identify some aspects of this task which should be addressed in future work.

We first note these models often correlate certain categories, namely *Prevention*, *Transmission* and *Forecasting*, much more closely than necessary. Even though these categories are semantically related and some overlap exists, the *Transmission* and *Forecasting* tags are predicted in conjunction with the *Prevention* tag much more frequently than what is observed in the labels as can be seen from the table in Appendix C. Future work should attempt to explicitly model correlation between categories to help the model recognize the particular cases in which labels should occur together. The first row in Table 4 shows a document labelled as *Forecasting* which is also incorrectly predicted with a *Prevention* label, exemplifying this issue.

Finally, we observe that models have trouble identifying discriminative sections of the document due to how much introductory content on the pandemic can be found in most articles. Future work should explicitly model the gap in relevance between introductory sections and crucial sentences such as thesis statements and article titles. In Table 4, the second and third examples would be more easily classified correctly if specific sentences were ignored while others attended to more thoroughly. This could also increase interpretability, facilitating analysis and further improvement.

## 5 Conclusion

We provide an analysis of document classification models on the LitCovid dataset for the COVID-19 literature. We determine that fine-tuning pre-trained language models yields the best performance on this task. We study the generalizability and data efficiency of these models and discuss some important issues to address in future work.

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

## A   Experimental Set-up

We split the LitCovid dataset into train, dev, test with the ratio 7:1:2.

We adopt micro-F1 and accuracy as our evaluation metrics, same as (Adhikari et al., 2019a). We use scikit-learn (Pedregosa et al., 2011) and Hedwig evaluation scripts to evaluate all the models. For preprocessing, tokenization and sentence segmentation, we use the NLTK library.

All the document classification models used in the paper, logistic regression [1] SVM [2] DocBERT [3], Reg-LSTM [4], Reg-LSTM [5], XML-CNN [6], Kim CNN [7] are run based on the implementations listed here and strictly followed their instructions. We used the following pre-trained language models, BioBERT [8], BERT base [9], BERT large [10] and the Longformer [11].

For reproducibility, we list all the key hyperparameters, the tuning bounds and the # of parameters for each model in Table A1. For the logistic regression and the SVM all hyperparameters used were default to scikit-learn and therefore are excluded from this table. For all models we train for a maximum of 30 epochs with a patience of 5. We used micro-F1 score for all hyperparameter tuning. All models were run on NVIDIA GeForce GTX 1080 GPUs.

---

[1] https://scikit-learn.org/stable/modules/generated/\sklearn.linear_model.LogisticRegression.html

[2] https://scikit-learn.org/stable/modules/generated/sklearn.svm.SVC.html

[3] https://github.com/castorini/hedwig/blob/master/models/bert

[4] https://github.com/castorini/hedwig/blob/master/models/reg_lstm

[5] https://github.com/castorini/hedwig/blob/master/models/reg_lstm

[6] https://github.com/castorini/hedwig/blob/master/models/xml_cnn

[7] https://github.com/castorini/hedwig/blob/master/models/kim_cnn

[8] https://huggingface.co/monologg/biobert_v1.1_pubmed

[9] https://huggingface.co/bert-base-uncased

[10] https://huggingface.co/bert-large-uncased

[11] https://github.com/allenai/longformer

| Model | Hyperparameters | Hyperparameter bounds | Number of Parameters |
|---|---|---|---|
| **Kim CNN** | batch size: 32
learning rate: 0.001
dropout: 0.1
mode: static
output channel: 100
word dimension: 300
embedding dimension: 300
epoch decay: 15
weight decay: 0 | batch size: (16, 32, 64)
learning rate: (0.01, 0.001, 0.0001)
dropout: (0.1, 0.5, 0.7) | 362,708 |
| **XML-CNN** | batch size: 32
learning rate: 0.001
dropout: 0.7
dynamic pool length: 8
mode: static
output channel: 100
word dimension: 300
embedding dimension: 300
epoch decay: 15
weight decay: 0
hidden bottleneck dimension: 512 | batch size: (32, 64)
learning rate: $(0.001, 0.0001, 1 \times 10^{-5})$
dropout: (0.1, 0.5, 0.7)
dynamic pool length: (8, 16, 32) | 1,653,716 |
| **LSTM** | batch size: 8
learning rate: 0.001
dropout: 0.1
hidden dimension: 512
mode: static
output channel: 100
word dimension: 300
embedding dimension: 300
number of layers: 1
epoch decay: 15
weight decay: 0
bidirectional: true
bottleneck layer: true
weight drop: 0.1
embedding dropout: 0.2
temporal averaging: 0.0
temporal activation regularization: 0.0
activation regularization: 0.0 | learning rate: (0.01, 0.001, 0.0001)
hidden dimension: (300, 512) | 3,342,344 |
| **LSTM$_{\text{Reg}}$** | batch size: 8
learning rate: 0.001
dropout: 0.5
hidden dimension: 300
temporal averaging: 0.99
mode: static
output channel: 100
word dimension: 300
embedding dimension: 300
number of layers: 1
epoch decay: 15
weight decay: 0
bidirectional: true
bottleneck layer: true
weight drop: 0.1
embedding dropout: 0.2
temporal activation regularization: 0.0
activation regularization: 0.0 | batch size: (8,16)
learning rate: (0.01, 0.001, 0.0001)
hidden dimension: (300, 512)
dropout: (0.5, 0.6) | 1,449,608 |
| **BERT$_{\text{base}}$** | learning rate: $2 \times 10^{-5}$
max sequence length: 512
batch size: 6
model: bert-base-uncased
warmup proportion: 0.1
gradient accumulation steps: 1 | learning rate: (0.001, 0.0001,
$2 \times 10^{-5}, 1 \times 10^{-6})$
maximum sequence length: (256, 512) | 110M |
| **BERT$_{\text{large}}$** | learning rate: $2 \times 10^{-5}$
max sequence length: 512
batch size: 2
model: bert-large-uncased
warmup proportion: 0.1
gradient accumulation steps: 1 | learning rate: (0.001, 0.0001,
$2 \times 10^{-5}, 1 \times 10^{-6})$
maximum sequence length: (256, 512) | 336M |
| **BioBERT** | learning rate: $2 \times 10^{-5}$
max sequence length: 512
batch size: 6
model: monologg/biobert_v1.1_pubmed
warmup proportion: 0.1
gradient accumulation steps: 1 | learning rate: (0.001, 0.0001,
$2 \times 10^{-5}, 1 \times 10^{-6}))$
maximum sequence length: (256, 512) | 108M |
| **Longformer** | learning rate: $2 \times 10^{-5}$
max sequence length: 1024
batch size: 3
model: longformer-base-4096
warmup proportion: 0.1
gradient accumulation steps: 1 | learning rate: (0.001, 0.0001,
$2 \times 10^{-5}, 1 \times 10^{-6}))$
maximum sequence length: (1024, 3584) | 148M |

Table A1: Hyperparameters, tuning bounds and number of parameters for each method.

# B  Performance by Category

| Category | Binary F1 Score | |
| --- | --- | --- |
| | LitCovid Dev | CORD-19 Set |
| **Prevention** | 88.2 ±0.2 | 65.8 ±2.9 |
| **Case Report** | 87.2 ±1.1 | 66.7 ±0.0 |
| **Treatment** | 81.5 ±0.5 | 60.5 ±4.2 |
| **Diagnosis** | 75.7 ±2.0 | 58.0 ±1.4 |
| **Mechanism** | 71.1 ±1.6 | 81.4 ±3.8 |
| **Forecasting** | 70.9 ±1.1 | 0.0 ±0.0 |
| **General** | 64.4 ±8.6 | 0.0 ±0.0 |
| **Transmission** | 48.3 ±3.7 | 52.0 ±11.0 |

Table A2: BioBERT Binary F1 scores per category on the *LitCovid* dev set and the *CORD-19* test set. Scores are given as *mean ± standard deviation* across three BioBERT training runs.

# C  Category Correlation

| Category | Full Label | Percentage of Docs with Category | |
| --- | --- | --- | --- |
| | | Label | Prediction |
| **Forecasting** | **Single Label** | 39.1 | 23.7 |
| | **+ Prevention** | 43.4 | 71.1 |
| **Transmission** | **Single Label** | 17.3 | 3.4 |
| | **+ Prevention** | 48.0 | 55.0 |

Table A3: This table shows how the Longformer model predicts (Forecasting & Prevention) and (Transmission & Prevention) much more frequently than can be found in the labels. The numbers are percentages of total number of documents with that category label.