# OpenReview forum: "Document Classification for COVID-19 Literature"
_aclweb.org/ACL/2020/Workshop/NLP-COVID — NLP-COVID-2020 Abstractonly_

### Official Review · AnonReviewer1 · 2020-07-01
**Review of Document Classification for COVID-19 Literature**

**Rating:** 6
**Confidence:** 3

**Review:**

This paper provides an empirical evaluation of several standard models' performances on a covid-19 related document classification task. Specifically, Covid-19 research papers are classified into one or more of eight subcategories based on their topic. Datasets for this task are constructed from LitCovid, and the Covid-19 Open Research Dataset. These datasets are shared online. Finetuned pre-trained language models are found to outperform more traditional models, such as SVM, CNN, or LSTM. In particular, BioBERT and Longformer are found to be most effective.

The paper introduces a multilabel document classification dataset which categorizes Covid19 related articles into one or more of eight subcategories. This dataset is distinct in that it represents a large body of work in the medical domain, united by a common subject. The data is drawn largely from LitCovid, with a supplemental test set drawn from the COVID-19 Open Research Dataset (CORD-19). The paper notes that models trained on the LitCovid data generalize poorly to the CORD-19 data, and theorizes that this is due to the models failing to ignore the overarching COVID-19 topic. It is unclear if this is the case, however, as, while CORD-19 includes papers related to other coronaviruses, COVID-19 is the overarching topic of both datasets.

This paper provides a thorough analysis of many different models' performance with thorough documentation. However, the efficacy of fine-tuned pretrained language models on document classification is well established in a large body of research. While this particular classification task is somewhat novel, demonstrating that finetuned pretrained language models are most effective upon it is a relatively limited contribution.

Pros
 - introduces a multilabel document classification task related to COVID-19
 - provides a dataset to support this classification task
 - provides a clear and thorough empirical evaluation of many different models' performances on this task

Cons
 - the empirical performance evaluation largely confirms trends shown by previous research
 - models trained on the provided dataset are shown to generalize poorly. The reasons for this are underexplored.

---

> ### Author Response · Authors · 2020-07-04
> **Response to Reviewer 1**
>
> We are grateful for the reviewer taking the time to review our work. We are glad that you found the proposed classification task to be novel and our empirical evaluation to be clear and complete. To address some of your concerns:
>
> > The paper notes that models trained on the LitCovid data generalize poorly to the CORD-19 data, and theorizes that this is due to the models failing to ignore the overarching COVID-19 topic. It is unclear if this is the case, however, as, while CORD-19 includes papers related to other coronaviruses, COVID-19 is the overarching topic of both datasets.
>
> Even though CORD-19 is seen as primarily a COVID-19 dataset, around half the articles in its first version were about older coronaviruses and other epidemics. Additionally, the 100 articles in our test set were drawn from the ones in CORD-19 but not in LitCovid, so their overarching theme is not COVID-19 but the general coronavirus family. In fact, this test set contains the term "COVID-19" in only one of the articles and is mostly unrelated to COVID-19. This setting aims to test whether the models have learned general signals discriminative for the categories or superficial signals specific to COVID-19. Nonetheless, we agree with the reviewer that it is worth further investigation to ascertain that the models' inability to ignore overarching COVID-19 information is the main reason why they fail to generalize.

---

### Official Review · AnonReviewer2 · 2020-07-03
**Overall a good exploration of previous document classification algorithms applied to Covid-19.**

**Rating:** 7
**Confidence:** 4

**Review:**

The paper presents an exploration of the main document multi-classification algorithms, training and developing on the LitCovid dataset and testing on a small manually annotated portion of the much larger CORD-19 dataset. The LitCovid dataset includes labels for each document to 8 categories with about 1/3 of documents including multi-labels. The significance lies in the usage of these multi-classification algorithms for researchers, clinicians, epidemiologists, and policy makers to find the most relevant articles in a timely manner given the urgency of the pandemic.

The paper is clear and well-written. The application of document classification algorithms to Covid-19 is novel, but the algorithms themselves are not.


Pros:
- An in-depth exploration of many different algorithms along with tuning for best performance.
- Contains both intrinsic and extrinsic evaluations.
- The discussion on data efficiency is quite interesting and can have a large impact on how much data is needed to annotate for future creation of gold standard datasets.
- An error analysis that suggest future work and updates.
- Important to be applying all research we have so far on these new datasets to help with Covid-19.



Cons:
- The manual annotation of the subset of CORD-19 seems to not be reliable as 34% of errors were annotation related.
- Guidelines and annotation process not provided for the CORD-19 set.
- The CORD-19 set is small and the performance does drop, suggesting that the models are not generalizable.
- No discussion of providing these models to LitCovid to help with the annotation labeling, even if it is a starting point.
- The accuracy is quite low for most algorithms and there is no discussion about this.
- Very little discussion on the very different distributions of the categories between LitCovid and CORD-19.

---

> ### Author Response · Authors · 2020-07-04
> **Response to Reviewer 2**
>
> We are grateful for the reviewer taking the time to read and review our paper. We are delighted that you found our work to be significant in addressing the current crisis and interesting. We address some concerns below:
>
> > The manual annotation of the subset of CORD-19 seems to not be reliable as 34% of errors were annotation related.
>
> The error analysis discussed in the paper was on LitCovid errors and not CORD-19 errors. Even though we did not explore the amount of annotation related errors in the CORD-19 test set we are confident that our annotation process was reliable due to the annotation process we discuss below.
>
> > Guidelines and annotation process not provided for the CORD-19 set.
>
> Since the categories are generally intuitive for biomedical experts annotation guidelines were not explicitly provided. The experts (all with years of biomedical research experience) were given a random sample of LitCovid articles for each category to familiarize themselves with the annotations.
> To come up with our annotations, as discussed in Section 2.2, we asked biomedical experts to label a small set of 100 articles found only in CORD-19. Each article was labelled independently by two annotators. For articles which received two different annotations (around 15%), a third, more senior annotator broke ties."
>
> > No discussion of providing these models to LitCovid to help with the annotation labeling, even if it is a starting point.
>
> We have reached out to the LitCovid team and would love it if these models would help make their annotation process easier, faster and more accurate.
>
> > The accuracy is quite low for most algorithms and there is no discussion about this.
>
> We agree that the accuracy scores are quite low for these models and believe that they can be partially explained by the same spurious label correlations we discussed in Section 4.3.
>
> > Very little discussion on the very different distributions of the categories between LitCovid and CORD-19.
>
> We appreciate your feedback, the discrepancy in label distributions between LitCovid and CORD-19 should be mentioned as a possible explanation for the generalization failure that should be further explored. To elaborate a bit more on where the distribution discrepancy originates. Many articles in CORD-19 were published before the COVID-19 pandemic and focus on general coronaviruses and previous epidemics. Therefore, a large portion of the CORD-19 articles study the disease mechanisms of other coronaviruses. However, since LitCovid is exclusively about the ongoing COVID-19 pandemic, there is a large number of articles discussing the transmission, prevention, and case reports, hence the distribution discrepancy.

---

### Official Review · AnonReviewer4 · 2020-07-05
**Study of automatic classification of LitCovid categories.**

**Rating:** 6
**Confidence:** 5

**Review:**

This article presents a comparison of methods for the classification of articles from the LitCovid collection. From all the evaluated methods, one based on BioBert performs better compared to other evaluated methods. The contribution of the paper is a comprehensive study of a broad range of methods to the multi-class classification of LitCovid. The classifiers are evaluated on a small set of papers from the CORD-19 collection. The article is clearly written and plenty of details. The authors provide the data and code, which would support reusability and reproducibility.

The performance on CORD-19 is significantly lower compared to the reported numbers for LitCovid. This might be due to several reasons, including the limited size of the CORD-19 articles used or the way manual annotation done by the authors might differ from the one performed in LitCovid.

The LitCovid collection already provides the classification, thus the classifier might have limited used unless a specific use case is specified in which this work would be used.

The title might be willing to include the specific collection on which the methods have been tested on, since the collections and the labels are provided by the LitCovid data set.

The article has been recently uploaded and currently has almost 28k PubMed articles. It might be relevant to mention when the documents were obtained. The authors mention 8k, thus probably it was obtained sometime in April.

In table 1, 2861 —> 2,861

---

> ### Author Response · Authors · 2020-07-05
> **Response to Reviewer 3**
>
> We are grateful for the reviewer taking the time to read and review our paper. We address some concerns below:
>
> > The article has been recently uploaded and currently has almost 28k PubMed articles. It might be relevant to mention when the documents were obtained. The authors mention 8k, thus probably it was obtained sometime in April.
>
> The version of the LitCovid dataset we evaluated on was obtained on May 20th and is available on our Github. It contained around 14,000 articles at the time out of which we used 8,000 that were either full articles or abstracts. The rest consisted of only article titles since their abstracts were not available on Pubmed. We are currently running experiments on the most up to date version of the dataset and include scripts to download the most up to date version of the LitCovid collection to make future evaluation straight forward.

---

### Decision · Program_Chairs · 2020-07-06

**Decision:**

Accept (Abstract only)

**Comment:**

Thank you for your submission and we are pleased to invite you to present this work at the workshop on Thursday (5:30-9:30pm PDT).

We look forward to hearing about your work.

Please plan on a 10 minute video presentation; pre-recorded is likely the best option.